# Impact of Protein Intake in Older Adults with Sarcopenia and Obesity: A Gut Microbiota Perspective

**DOI:** 10.3390/nu12082285

**Published:** 2020-07-30

**Authors:** Konstantinos Prokopidis, Mavil May Cervo, Anoohya Gandham, David Scott

**Affiliations:** 1Department of Digestion, Absorption and Reproduction, Faculty of Medicine, Imperial College London, White City, London W12 0NN, UK; 2Department of Medicine, School of Clinical Sciences at Monash Health, Monash University, 3168 Clayton, Victoria, Australia; mavil.cervo@monash.edu (M.M.C.); anoohya.gandham@monash.edu (A.G.); d.scott@deakin.edu.au (D.S.); 3Institute for Physical Activity and Nutrition, School of Exercise and Nutrition Sciences, Deakin University, 3125 Burwood, Victoria, Australia; 4Department of Medicine and Australian Institute of Musculoskeletal Science, Melbourne Medical School–Western Campus, The University of Melbourne, 3021 St Albans, Victoria, Australia

**Keywords:** sarcopenia, obesity, gut microbiota, sarcopenic obesity, skeletal muscle, protein, older adults, short-chain fatty acids

## Abstract

The continuous population increase of older adults with metabolic diseases may contribute to increased prevalence of sarcopenia and obesity and requires advocacy of optimal nutrition treatments to combat their deleterious outcomes. Sarcopenic obesity, characterized by age-induced skeletal-muscle atrophy and increased adiposity, may accelerate functional decline and increase the risk of disability and mortality. In this review, we explore the influence of dietary protein on the gut microbiome and its impact on sarcopenia and obesity. Given the associations between red meat proteins and altered gut microbiota, a combination of plant and animal-based proteins are deemed favorable for gut microbiota eubiosis and muscle-protein synthesis. Additionally, high-protein diets with elevated essential amino-acid concentrations, alongside increased dietary fiber intake, may promote gut microbiota eubiosis, given the metabolic effects derived from short-chain fatty-acid and branched-chain fatty-acid production. In conclusion, a greater abundance of specific gut bacteria associated with increased satiation, protein synthesis, and overall metabolic health may be driven by protein and fiber consumption. This could counteract the development of sarcopenia and obesity and, therefore, represent a novel approach for dietary recommendations based on the gut microbiota profile. However, more human trials utilizing advanced metabolomic techniques to investigate the microbiome and its relationship with macronutrient intake, especially protein, are warranted.

## 1. Introduction

Older population numbers are expected to rise dramatically over the upcoming decades across the globe. By 2050, the worldwide population of those aged over 65 years is projected to increase by approximately 10%, reaching 2.1 billion [1]. It is estimated that metabolic syndrome is prevalent in 12–26% and 12–37% of the European and Asian population, respectively [2]. Metabolic syndrome is outlined by skeletal-muscle-insulin resistance, hypertension, hyperlipidemia, and abdominal obesity [3]. In aging populations, prolonged malnutrition is linked to metabolic syndrome-related diseases, including sarcopenia and obesity, which may emerge from increased body fat, proinflammatory cytokines, oxidative stress, mitochondrial dysfunction, hormonal changes, and insulin resistance [4]. Sarcopenia is characterized by an age-related loss of skeletal-muscle mass and function, beginning in our 30s and 40s and being highly prevalent entering our sixth decade [5]. Sarcopenia is often accompanied by loss of balance, increased morbidity, and frailty due to deterioration of muscle fibers [6], restricting the ability of individuals to remain physically active, leading to subsequent disabilities and dependency [7,8]. The onset of sarcopenia includes inflammatory responses, oxidative stress, reduced energy expenditure, and decreased appetite [9,10]. Although several definitions of sarcopenia are currently available, including the recently revised European Working Group on Sarcopenia in Older People (EWGSOP) definition, the lack of a universal consensus on definition limits comparison between studies and amongst various population groups [11]. The prevalence of sarcopenia is largely influenced by the definition used, and the lack of a consensus also makes it difficult for clinicians to identify older adults at risk of this condition [12]. In addition, approximately 30% of the population globally is classified as overweight or obese, highlighting an additional public health and financial burden [13]. Obesity is caused by chronic positive energy balance, leading to increased proinflammatory cytokine expression, adipocyte and immune dysfunction, and insulin resistance, which contribute to a range of metabolic diseases [14], including type 2 diabetes and cancer [15]. A body mass index (BMI) > 30 kg/m^2^ and a waist circumference over 102 cm and 88 cm for men and women, respectively, have been used as measures identifying obese populations [16]. In the United States, 38% of males and 39% of females over the age of 60 are considered obese [17], whereas the proportion of adults above a BMI of 25 kg/m^2^ increased by 8% in the period of 1980–2013 [18].

Gut microbiota dysbiosis is linked to age-related systemic inflammation, leading to impaired muscle function and increased proinflammatory cytokines, which are associated with higher risk of obesity [19]. Gut microbiota dysregulation may promote the onset of sarcopenia and obesity through myostatin and atrogin-1 expression [20] and dysfunctional signaling between the enteric nervous system and the brain, respectively [21], imposing a negative impact on muscle mass and appetite. Nutrition may be a pivotal contributor to gut microbiota regulation, although different macronutrients promote distinct properties on the microbiome [22]. Short-chain fatty-acid (SCFA) production from dietary fiber is suggested as a prominent mediator of the gut microbiota through bacterial fermentation in the gastrointestinal tract, interacting with several gut–brain signaling pathways [23]. In addition, considering the skeletal muscle anabolic effects of dietary protein, induction of increased protein intakes above the recommended dietary allowance (RDA) 0.8 g/kg/day is considered a valuable tool to counteract the gradual muscle loss and increased appetite for food [24]—a main characteristic of sarcopenic obesity. However, questions have been raised regarding the effects of dietary proteins on the gut microbiota and the health impact induced by their bioactive end-products [25].

In this review, we discuss how dietary protein may influence the gut microbiota ecosystem and its potential role in sarcopenia and obesity. We further aim to provide perspectives on novel future dietary recommendations, focusing primarily on the potential anabolic-induced effects that emerge from elevated protein and fiber consumption.

## 2. Sarcopenia and Obesity

Sarcopenia is accompanied by a greater incidence of comorbidities, including type 2 diabetes and obesity [26,27,28], which are indicative of the devastating effects derived by systemic inflammation and insulin resistance [29]. In obese older individuals, the catabolic-induced aging and obesogenic environments associated with sedentary behavior and malnutrition may lead to gradual fat accumulation in adipose tissue and simultaneous impaired skeletal-muscle atrophy, leading to a condition known as sarcopenic obesity [30,31]. Sarcopenic obese older adults are often perceived as “fat–frail” due to the weakness and poor mobility associated with sarcopenia potentially being exacerbated by excess bodyweight from obesity [32]. Recent evidence indicates that the coexistence of sarcopenia and obesity is associated with adverse musculoskeletal outcomes [33], accelerated functional decline, and worse disability compared with those with sarcopenia or obesity alone, or neither sarcopenia nor obesity [34]. This may lead to increased incidence of falls [35,36] and, combined with the apparent loss of any protective effect of obesity for fractures [37], sarcopenic obesity may perpetuate disability and poor quality of life [38]. Recently, there has been conjecture over whether high fat mass does in fact increase the risk of functional decline in older age beyond the effects of sarcopenia alone [39]. Nonetheless, urgency is required for recommendations on proper diagnosis and management of sarcopenic obesity, because each component is a potentially modifiable risk factor for poor metabolic health and mortality risk [40]. Like sarcopenia, clinical recognition of sarcopenic obesity among older patients is limited due to a lack of consensus on definitions, but presentations related to this condition will increase substantially with concurrent obesity epidemics and aging of populations internationally [31]. Currently, dietary recommendations for sarcopenic obesity involve high-protein, hypocaloric diets intended to improve body composition [31,41]. Considering that physical frailty may be a pivotal obstacle to aerobic and resistance exercise performance, greater attention should be paid to multiple domains of nutritional strategies. A developing field in the role of optimal dietary patterns for sarcopenia and obesity management is characterizing the ideal eubiosis of the gut microbiota ecosystem [42,43].

## 3. Gut Microbiota and Immune and Metabolic Homeostasis

### 3.1. Types and Functions of Gut Microbiota

The human gut is colonized by numerous microorganisms (e.g., bacteria, eukaryotic microbes, viruses, fungi, archaea), which are considered a contributor to a range of physiological functions, such as strengthening the gut integrity [44], regulating host immunity [45], and maintaining metabolic health [46]. A recent study showed that the ratio of human to bacterial cells is almost 1:1 [47]. The major phylogenetic types within the digestive tract consist of Bacteroidetes (*Porphytomonas*, *Prevotella*), Firmicutes (*Ruminococcus*, *Clostridium*, and *Eubacteria*), Proteobacteria with minor groups of Actinobacteria (*Bifidobacterium*), Acidobacteria, Fusobacteria, and Verrumicrobia [48]. Firmicutes (*Lactobacillus*, *Veillonella*) and Proteobacteria (*Helicobacter*) are dominant in the proximal gut, whereas Firmicutes (*Lachnospiraceae*) and Bacteroidetes are observed in the colon [49,50]. The different types of gut microbiota are illustrated extensively in Figure 1.

Due to the antimicrobial effects of gastric and bile-acid secretion, a relatively low number of microbiota are present in the stomach and the proximal small intestine [52]. The composition and properties of the microbiome may depend on their occupation, as microbial populations at the mucosa surface and the lumen interact with the host immune system and the metabolic effects of food, respectively [53]. The gut microbiome has a critical role in the immune system through the prevention of pathogen colonization, stimulation of immunoglobulin A production, upregulation of anti-inflammatory cytokines, and T cell regulation [53]. For instance, *Faecalibacterium prausnitzii* and *Bifidobacterium infantis* may result in the production of the anti-inflammatory cytokine interleukin-10 and regulate T cell activation against the pathogen-stimulated NF-κB inflammatory pathway [54]. Other species may additionally induce lower inflammation levels by interleukin-17 expression, assisting host immunity in the protection of detrimental pathogens [55]. Furthermore, the gut microbiome is critical in the de novo synthesis of essential vitamins, such as vitamin B12, folate, vitamin K, nicotinic acid, pyridoxine, and others [46], as well as bile acids [56]. Alteration of the co-metabolism of bile acids and vitamins has been associated with the development of metabolic diseases, such as obesity and type 2 diabetes [57]. A catalogue of the functional capacity of the human gut microbiome identified 9,879,896 genes wherein country-specific microbial signatures were found suggesting that gut microbiota composition is affected by multiple factors, such as host genetics, diet, health status, aging [58,59,60], and antibiotic administration [61].

### 3.2. Gut Ecosystem and Metabolic Health

The gut microbiota has a substantial impact on the regulation of metabolic processes, including nutrient and amino-acid metabolism [62,63]. Alterations in microbial composition may occur within 24 h following a change in dietary patterns [64], although greater changes may require a long-term adherence [65]. Microbial alterations are associated with multiple diseases, such as cancer, sarcopenia, obesity, and cardiovascular diseases [48,66,67]. An association between greater medication use with increased incidence of physical frailty and sarcopenia has been observed, which may be partly explained by the potential impact of polypharmacy on microbiota composition [68]. Indeed, distinct differences in the microbiota of healthy and frail older adults may partially explain the onset of frailty and sarcopenia [69] through effects on the mTOR signaling pathway—a major proponent of muscle protein synthesis (MPS) [70]. Moreover, dysfunctional musculoskeletal health has been suggested to be modulated by proinflammatory responses occurring in the microbiome [71]. Inflammatory responses associated with aging may induce microbial alterations influenced by pathogens, malnutrition, and lower overall lifestyle quality, leading to intestinal mucosa permeability [25,72,73]. Intestinal permeability initiates altered gut microbial composition (dysbiosis) and increases the levels of proinflammatory cytokines, such as interleukin-6 (IL-6) and tumor necrosis factor-α (TNF-α) [74,75]. Gut microbiota dysbiosis has a critical role in the gut–muscle axis through mitochondrial dysfunction, affecting skeletal-muscle metabolism further [76,77,78,79]. This may be perpetuated by reactive-oxygen-species (ROS) production in the elderly [80], which activate the NF-κB signaling pathway [81]—an activator of IL-6 and TNF-α release [76]. In obese groups, microbial diversity is significantly lower compared to lean population groups [82]. Microbial diversity including *Bifidobacterium, Lactobacillus, Akkermansia, Fecalibacterium, Eubacterium, Roseburia, Ruminococcus*, and *Blautia* species are considered beneficial for metabolic health in the elderly [83,84]. For instance, in leukemia-disordered mice, *Lactobacillus*-species restoration reduced muscle atrophy, which is also correlated with a decrease in various proinflammatory cytokines [85]. On the contrary, species, such as *Clostridium, Enterobacter, Enterococcus*, and *Ruminococcus* are associated with altered energy balance and greater risk of obesity [51]. Multiple studies have linked dietary patterns with changes in the gut ecosystem, displaying a distinct role of macronutrients and their impact on the gut microbiota environment [64,86]. However, the complexity in trying to understand the metabolic effects of gut microbiota and their relationship with detrimental health conditions is challenging [87]. Most studies have been performed in mice, which share 95% similar gut microbiota functionality with humans [88].

Moreover, alterations in gut microbiota composition are observed more commonly in people over the age of 65 compared to younger adults [89,90]. Higher microbial diversity from increased *Bifidobacterium* and *Lactobacillus* may promote more efficient nutrient absorption and amino-acid synthesis, whereas low microbial diversity is associated with excess nutrient uptake and storage and increased Firmicutes to Bacteroidetes ratio [91,92]. A greater abundance of Firmicutes relative to Bacteroides concentration is characterized in the microbiome of obese and insulin resistant humans and animals [93,94]. Interestingly, decreased *Bifidobacterium* levels are detected during aging, which is linked to increased circulating levels of lipopolysaccharide (LPS). Elevated levels of the endotoxin LPS are marked in obese and diabetic individuals, leading to gut microbiota dysbiosis, skeletal-muscle-insulin resistance and increased gut permeability [95,96,97,98,99]. LPS is a marker of endotoxemia, which promotes skeletal-muscle-insulin resistance by proinflammatory cytokine expression of TNF-α, interleukin-1 (IL-1), interleukin-2 (IL-2), and IL-6 [100,101]. This highlights the potential of age-related decrements in gut microbiota (e.g., *Bifidobacterium*) to influence the development of sarcopenic obesity through decreased glucose tolerance in the skeletal muscle [95,100,102,103,104]. Furthermore, inflammatory cytokines induced during aging by enhanced LPS levels have demonstrated suppressed protein synthesis via muscle-protein synthesis and muscle-protein breakdown (MPS: MPB) imbalance, leading to reduced muscle mass and physical function [105,106,107,108,109]. In older adults, serum LPS and gene expression of its receptor, Toll-like receptor-4 (TLR4), is linked to lower insulin sensitivity compared to younger groups, showing that age-related LPS levels may increase the incidence of insulin resistance during aging [103,110]. Therefore, reductions in microbial diversity and functionality of the host may influence the functionality of several organs, including the skeletal muscle [109,111,112]. However, questions have been raised as to whether altered microbial diversity is caused by aging or if the microbiome is responsible for the consequences derived by aging [113,114].

## 4. Sarcopenic Obesity: A Case for Protein and Gut Microbiota

### 4.1. Dietary Protein and Gut Microbiota

Protein is the dominant macronutrient in weight-loss strategies combatting sarcopenic obesity, given its appetite-suppressive effects [115] and anabolic effects on maintaining MPS above MPB [116]. The G protein-coupled receptors (GPCRs) located in the L- and G-cells of the colon and the small intestine, respectively, modulate glucagon-like peptide-1 (GLP-1) and peptide YY (PYY) secretion through amino-acid sensing, impeding the stimulation of food-intake regulatory effects occurring in the gut–brain axis [117,118,119]. In addition, satiation is further augmented by cholecystokinin (CCK) release, which is stimulated by protein consumption [120]. Studies have confirmed these effects compared to dietary carbohydrates and fats consumption [121], which may be attributed to the regulation of leptin and ghrelin secretion [122,123,124,125]. It is worth mentioning that, appetite-induced responses driven by signals between the gut microbiota and dietary protein may be determined by amino-acid composition [126] and, particularly, essential amino acids (i.e., leucine) [127,128]. Although dietary protein is established as a competent appetite regulator, its satiating and anabolic effects in older adults may be alleviated following lower protein diets [129]. The current RDA for protein at 0.8 g/kg/day may be insufficient for older adults due to their inability to absorb and utilize protein to the same extent compared to younger individuals [106,108,130]. Recommendations for leucine consumption—around 3–4 g per meal, which equates to 25–30 g of high quality protein and 1.0–1.6 g/kg/day distributed into 3–4 daily meals—aim to promote greater MPS stimulation in older adults [131,132,133,134]. Dietary protein is the gut microbiota’s primary source of amino acids, which can be used for protein synthesis and energy metabolism [135]. Currently, there is a controversy around the gut microbiota and high-protein diets in metabolic health and disease during aging. Microbiome changes are suggested to be engaged directly or indirectly in several mechanisms of age-related anabolic resistance, which may explain the necessity of greater protein intake in aging populations [136]. Anabolic resistance is associated with reduced gene expression in proteins involved in MPS, impaired protein absorption and digestion, loss of skeletal-muscle stem cells, and decreased amino-acid transportation in the skeletal muscle [19,137,138]. Furthermore, malnutrition and a sedentary lifestyle are proponents of anabolic resistance, increasing gradually with aging [116]. This raises questions on the impact gut microbiota may have on metabolic diseases, including the onset of sarcopenia and obesity [69].

Most protein is digested and absorbed efficiently in the small intestine by pancreatic enzymes and peptidases used by enterocytes, although approximately 10% of proteins that pass through the small intestine may not be completely digested [139]. Going to the large intestine for further proteolysis by the colonic microbiota, amino acids are not absorbed by the colonocytes as efficiently and some metabolites may be used for metabolic or waste products [140,141]. The transit time and microbiota concentration is greater in the large than the small intestine, with bacterial proteases and peptidases breaking down endogenous and dietary proteins to peptides and amino acids [142]. The undigested proteins and peptides that reach the colon influence gut microbiota production and composition, contributing to large amounts of indigestible products [143,144,145]. Regarding this, as protein consumption is increased, the amount of proteins reaching the colon is increasing correspondingly, leading to numerous and diverse bacterial metabolite production (e.g., hydrogen sulfide, branched-chain fatty acids (BCFAs), SCFAs, polyamines, ammonia, methane, aromatic compounds, nitric oxide, tyramine, tryptamine, phenethylamine, serotonin, histamine, and others) in the gastrointestinal tract [141,146,147]. Some of these metabolic products are detrimental for metabolic health and are associated with chronic inflammation and several diseases (e.g., inflammatory bowel disease, colorectal cancer). However, to date, there is no causal link in humans, taking into account the absence of long-term experimental trials on high protein diets and the gut microbiome and their multifaceted relationships [148,149].

Multiple human and animal studies have linked increased branched-chain amino acids (BCAAs) with insulin resistance and type 2 diabetes in obese groups [150,151,152,153]. However, increased SCFA consumption may alleviate the hyperglycemic responses that are occurring in obese and type 2 diabetics, characterized by elevated amino-acid concentrations [154]. In accordance, undigested amino acids by the colonic epithelium may be used by the host through BCFA and SCFA activity to regulate protein homeostasis and energy production by muscle cells [141,142,155,156]. BCAA deamination leading to BCFA production is a marker of colonic fermentation developed by protein consumption [147]. The conversion of BCAA valine, leucine, and isoleucine to isobutyrate, isovalerate, and 2-methylbutyrate, respectively [157], may contribute to approximately 5% of the total SCFA production [158]. This evidence indicates that the composition and concentration of amino acids may play a pivotal role in the proteolytic fermentation by the gut microbiota in the small intestine, which influences amino-acid homeostasis [159,160]. Therefore, it is recommended that high-protein diets should be carefully designed, considering the levels of protein fermentation by the gut microbiota and the amount of protein entering the large intestine [139,145].

### 4.2. Protein Sources, Amino Acids, and Gut Microbiota Species

It is suggested that protein sources and amino-acid balance may influence gut microbial diversity. For instance, plant proteins are associated with greater *Bifidobacterium*, *Roseburia*, *Ruminococcus bromii*, *Lactobacillus*, and *Roseburia* content [161], as opposed to *Bacteroides*, *Alistipes*, *Bilophila*, and *Clostridium perfrigens*, found primarily in animal proteins [64,162]. A greater abundance of *Bacteroidetes*, *Bifidobacterium*, and decreased serum LPS levels compared to meat, dairy, and casein-protein consumption have been associated with soy-protein intake [163,164]. Furthermore, increased *Bifidobacteria and Lactobacilli*, which are linked to decreased diet-induced obesity and improved insulin sensitivity have been further supported by soy-protein consumption [165,166,167]. Likewise, increased bile-acid transformation, GLP-1 secretion, elevated *Lactobacillus* and *Bifidobacterium* levels, and reduced Firmicutes have been reported following soybean, mungbean, and buckwheat proteins [168,169]. In addition, Bifidobacteria-fermented whey and cheese protein have expressed decreased populations of *Bacteroides fragilis* and *Clostridium perfingens*, increased acetate production, and greater *Lactobacillus* and *Bifidobacterium* diversity [170,171,172]. Moreover, certain *Lactobacillus* and *Bifidobacteria* species have been associated with increased muscle strength, weight loss, and reduced obesogenic environments in humans and rodents [173,174,175,176]. This may be attributed to whey protein’s abundance in *Lactobacillus* and *Bifidobacteria*, as reported in rodent studies [177,178,179]. Regarding this, higher *Lactobacillus* abundance has been demonstrated from white-meat-protein consumption, while supplementation of *Lactobacillus plantarum* has resulted in increased muscle mass in mice [175,177,179,180,181]. Examples of gut microbiota studies incorporating different protein types and sources, and their metabolic effects, are depicted in Table 1.

Furthermore, greater SCFA content and reduced Proteobacteria (*Helicobacter*) have been found in mice supplemented with seafood protein, which is characterized by increased taurine levels [184,186]. Additionally, higher *Bifidobacterium*, *Lactobacillus*, and SCFA production have been observed by high pea-protein intake, which suppresses the secretion of inflammatory cytokines, IL-6 and TNF-α, and improves interleukin-10 (IL-10) expression and glucose homeostasis [187,188,189,190]. Conversely, heterocyclic amines and glycan derived from red meat may promote inflammation in gut health due to higher concentration of *Bacteroides* and *Fusobacterium* and lower levels of *Lactobacillus* and *Roseburia*, which are also linked with lower anti-inflammatory responses and increased incidence of type 2 diabetes [191,192,193]. Accordingly, L-carnitine present in red meat can be metabolized to trimethylamine oxide (TMAO), which is associated with an increased incidence of atherosclerosis [194] and obesity [195]. This may not be compatible, however, with studies yielding higher circulating TMAO following consumption of seafood and fish products, known to be cardio-protective, compared with eggs and red meat proteins [196,197]. In addition, a high mixed whey–beef protein supplement for 70 days given in endurance athletes, reduced *Roseburia*, *Bifidobacterium longum*, and *Blautia* and increased *Bacteroidetes* species compared to the control group receiving maltodextrin [185]. However, in another study, high-protein beef supplementation in germ-free vs. unaltered microbiome mice depicted grip strength improvement in both groups [198], questioning the effects of high-protein beef administration in mice with different microbial composition. Likewise, the mice colonized with gut microbiota from older adults with increased functionality displayed enhanced muscle strength, exhibiting a greater abundance of the *Prevotellaceae* family compared to the lower-functionality older-adult donors [198]. Microbiota transplants from pathogen-free mice have also shown reduced skeletal-muscle atrophy and mitochondria dysfunction markers than germ-free mice; more notably increased serum choline levels and neuromuscular junction proteins, Rapsyn and Lrp4 [199]. Overall, white-meat protein (chicken, fish) demonstrates positive outcomes for the host vs. red-meat protein (beef, pork) due to increased abundance of *Lactobacillus* [200]. It is worth noting that, certain pro- and prebiotic products contain *Bifidobacterium* and *Lactobacillus* species for their regulatory effects on the microbiome, bone, and muscle health, which are crucial against frailty phenotypes in older individuals [201,202,203]. Interestingly, favorable microbial composition has been displayed by plant proteins compared to white meat and, to a greater extent, red meat proteins, partly due to a higher proportion of SCFA-producing bacteria. However, results should be treated with caution considering the lack of experimental human studies [204,205,206]. Moreover, the variety of amino acids from different protein types available to the intestinal bacteria may regulate whole-body amino acid metabolism and protein utilization [207]. Metabolism of serine, aspartame, and alanine are regulated in the small intestine by L-glutamine, while phenylalanine, tyrosine, and tryptophan are involved in species found in *Clostridium bartlettii*, *Eubacterium hallii*, and *Bacteroides* [208,209]. In addition, lysine, glutamate, glycine, ornithine, aspartame, and threonine may contribute to acetate metabolism, whereas lysine, glutamate, and threonine to butyrate synthesis [63,141]. This indicates that amino acids are crucial for SCFA synthesis showing great versatility regarding the production and synthesis of different SCFAs. Furthermore, delayed age-associated microbiota changes in mice have been observed by BCAA consumption, displaying a greater abundance of *Bifidobacterium* and *Akkermansia* [210], which improve glucose homeostasis and insulin sensitivity [182,211,212]. Although beneficial and deleterious effects from protein consumption have been identified, the health effects of amino acids on the metabolic human phenotype interfered with the gut microbiota that depend on protein digestibility and absorption are yet to be fully understood.

### 4.3. Protein Utilization in the Gut: The Role of Dietary Fiber

Short-chain fatty-acid production is primarily derived from non-digestible-carbohydrate (i.e., dietary fiber) consumption [213] during colonic bacterial fermentation [214]. These fermented products include acetate, propionate, and butyrate, approximately in a 60:20:20 ratio, respectively [215], and the two non-digestible carbohydrate categories are soluble (pectin, guar gum, psyllium, inulin) and insoluble fiber (cellulose, hemicellulose, lignin) [216]. The recommended intake for dietary fiber is deemed to be country-specific. For example, in the UK, the dietary target is settled at 30 g/day, while in Australia it is at 28 g and 38 g for women and men, respectively. However, in both countries, the majority of the population does not meet the suggested fiber intake [217,218]. Dietary fiber may induce an array of metabolic effects [219], including reduced systemic inflammation by regulating cytokine expression, primarily interleukin-18, improved fat oxidation and insulin sensitivity [220,221,222,223]. The emerging role of SCFAs on skeletal-muscle metabolism and function was recently reviewed [224]. In addition, increased leucine levels are correlated with enhanced butyrate and propionate concentrations in pigs, displaying a greater abundance of Actinobacteria species and body-fat loss [183]. This may indicate that the microbiome may play a pivotal role in mTOR activation and leucine metabolism in the intestinal epithelial cells, possessing a propitious role in metabolic health [86]. Likewise, amino acids, such as tryptophan, alanine, and phenylalanine may also impact satiety and gut motility through GLP-1, PYY, and serotonin modulation from the intestinal enteroendocrine L cells [225,226,227,228,229,230,231]. SCFAs may act as substrates in several tissues for GPCRs, stimulating GLP-1 and PYY release, delaying gastric emptying and reducing appetite and food intake [227,232,233]. Similarly, it has been shown that propionate attenuates reward-based eating behavior via striatal pathways [234], which are linked to hyperpalatable food consumption, known for its high-calorie content and association with obesogenic environments [235]. Hence, the potent anabolic effects and peripherally appetite-induced responses of SCFAs and amino acids, particularly BCAAs, could reduce the risk of anabolic resistance in parallel with greater adiposity, which are precursors of sarcopenic obesity.

Furthermore, the influence of protein fermentation in the gut is primarily regulated by substrate utilization and transit time. It is suggested that the ratio of carbohydrates and protein consumed impacts the protein utilized by the microbiome. Following a 2-week high-protein diet (1.5–2.2 g/kg/day) in older women, microbial composition remained unaltered in the absence of added probiotic formulation [236]. The suppressed butyrate-producing populations of *Roseburia* and *Anaerostipes* were lower than the probiotic added groups and no changes were seen regarding *Eubacterium* and *Ruminococcus*, possibly due to the higher fiber intake of the plant-based products. Accordingly, a synbiotic-used probiotic component expressed increased *Bifidobacterium* and *Lactobacillus* during a weight-loss program containing a high-protein/low carbohydrate diet compared to the placebo group, indicating the potential requirement of probiotic supplementation for improved microbial richness [237]. Moreover, high-protein/low-carbohydrate diets in obese subjects showed reduced butyrate-producing bacteria and decreased levels of *Roseburia* and *Eubacterium rectale*, displaying a dose–response relationship, as carbohydrate intake was decreasing [148,238,239]. Likewise, decreased *Eubacterium rectale* and *Bifidobacteria* have been observed during weight-loss strategies in overweight and obese individuals in which further resistant-starch-carbohydrates reductions were common [240,241].

Altered microbial composition by increased *Bacteroides* and *Dorea* and reduced *Faecalibacterium* species have also been reported in elite race walkers following a 2.2 g/kg/day diet containing <50 g carbohydrates, although their training and body composition status are confounding factors [242]. In animals, decreased butyrate-producing bacteria from *Roseburia genera*, *Faecalibacterium*, and *Clostridium XIVa*, as well as increased Firmicutes, have been confirmed by high-protein consumption as opposed to moderate protein intakes [243,244]. The bacterial and metabolic effects of high-protein and low-carbohydrate (low-CHO) diets in humans and animals are summarized in Table 2.

Overall, complex carbohydrate availability may lower protein fermentation, leading to a greater number of nitrogenous substrates intended to promote muscle anabolism. Therefore, recommendations on the ratio and source of dietary protein [245] and carbohydrate consumption focusing on dietary fiber would be pivotal for skeletal-muscle and metabolic health through the impact of the metabolites generated in the large and small intestine [160].

## 5. Conclusions

The above sections illustrate the necessity of evaluating appropriate dietary components with the inclusion of both animal and plant-based food sources to optimize certain levels of gut microbiota species within high-protein diets. Regulating the quantity and source of food products may serve as a critical component for the control of protein and carbohydrate-fermenting bacteria, which could greatly influence various metabolic pathways. Given that different high-protein foods contain a varied micro- and macronutrient profile with notable distinctions among legumes, dairy, red and white meat, the microbial environment responsible for benefits and drawbacks in sarcopenic obesity cannot be accredited to a specific protein origin.

Consequently, the amount and source of protein combined with other lifestyle factors need to be defined in regards to microbial diversity, which further supports a personalized overall macronutrient approach considering individual variation in microbial composition. Gender, ethnicity, medical history, medication use, physical activity, genetics, local environment, and diet may all contribute significantly to different microbiota composition in older adults. Designing novel dietary patterns by examining these factors carefully in relation to specific microbial species may lead to a reduced incidence of obesity and improvements in skeletal-muscle-insulin sensitivity, as well as counteract sarcopenia and obesity during weight loss interventions using high-protein and fiber-rich diets. Therefore, biomarkers identifying dietary protein’s digestive capacity could aid older groups in monitoring their optimal daily protein intake and would minimise inappropriate amino acids reaching the colon and provide greater caloric availability for increased consumption of other vital macronutrients, such as dietary fiber, to optimize gut microbial eubiosis. Furthermore, fortification of plant-based products with essential amino acids and the addition of host-protective bacteria in high-protein animal products (e.g., kefir) could provide optimal MPS synthesis and microbial diversity. However, the overall quality of foods and supplements, alongside their protein content should also be evaluated considering the gut microbial changes that may occur throughout food processing. Moreover, the lack of long-term experimental human trials, more effective gene-sequencing methods for bacterial identification, and microbial exploration in several organs of the gastrointestinal tract present challenges in predicting the appropriate timing during which microbiome testing should be performed to identify microbiome changes occurring over time. At present, more human studies investigating the microbiome of sarcopenic obese groups are warranted. Focusing on the establishment and promotion of novel dietary protein and fiber RDAs, and aiming for greater BCFA and SCFA abundance, respectively, could provide various metabolic benefits in sarcopenic and obese individuals before we move towards a more personalized dietary approach through advanced microbiome metabolomic techniques.

## Figures and Tables

**Figure 1 nutrients-12-02285-f001:**
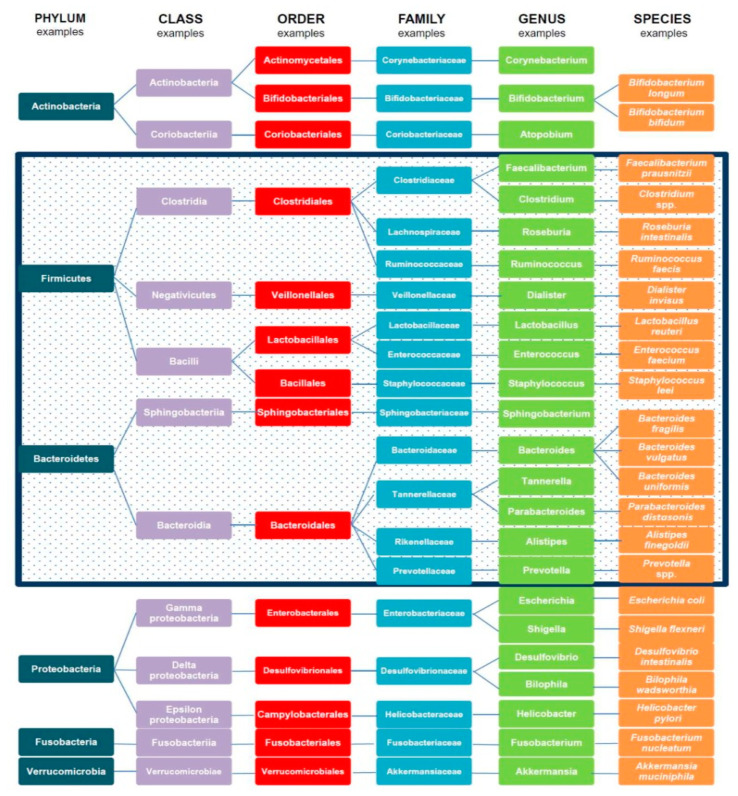
Bacterial types in the microbiome. Firmicutes and Bacteroidetes in the highlighted box represent approximately 90% of the total gut microbiota. Note. Adapted from [51].

**Table 1 nutrients-12-02285-t001:** Metabolic effects of protein supplementation and gut microbiota interaction in selected animal and human studies.

Study Type	Bacterial Type	Metabolic Effects	References
**Male mice supplemented with BCAAs**	Akkermansia↑ Bifidobacterium↑ Bacteroidetes↑ Proteobacteria↓ Actinobacteria↓ Firmicutes↓	Weight change↔ LPS↓	[182]
**Pigs supplemented with Leucine**	Actinobacteria↑ Lactobacillus↑ Firmicutes↑	Fat oxidation↑ SCFAs↑ LDL cholesterol↓ Fat mass↓	[183]
**Mice supplemented with Taurine**	Proteobacteria↑ (Helicobacter)	SCFAs↑ LPS↓	[184]
**Mice fed with whey isolate vs. casein for 21 weeks**	Lactobacillus↑ Clostridium↓	Lean mass↑ Leptin↓ Fat mass↓	[179]
**Mice fed with buckwheat vs. casein protein diets for 6 weeks**	Lactobacillus↑ Bifidobacterium↑ Enterococcus↑ Clostridium↑ Bacteroides↓	SCFAs↑ TNF-α↓ IL-6↓ LPS↓	[169]
**Mice fed with mungbean protein isolate for 4 weeks**	Bacteroidetes↑ Firmicutes↓	GLP-1↑ PYY↔ Insulin↔	[168]
**Hamsters supplemented with soy vs. milk protein**	Bifidobacteria↑ Clostridiales↑ Bacteroidetes↓ Proteobacteria↓ (Helicobacter)	LDL cholesterol↓ HbA1c↓	[164]
**Endurance athletes supplemented with whey isolate + beef hydrolysate for 10 weeks**	Bacteroidetes↑ Bifidobacterium longum↓ Roseburia↓ Blautia↓	-	[185]
**Healthy humans supplemented with Bifidobacterium breve C50-fermented whey protein for 7 days**	Bifidobacteria↑ Bacteroides fragilis↓ Clostridium perfringens↓	Β-galactosidase↑ Nitroreductase↓ β-glucuronidase↓	[170]

↓ indicates decrease; ↑ indicates increase; ↔ indicates no change. SCFAs: Short-chain fatty acids, LDL: Low-density lipoprotein, TNF-α: Tumor necrosis factor alpha, IL-6: Interleukin-6, PYY: Peptide YY, HbA1c: Glycated hemoglobin.

**Table 2 nutrients-12-02285-t002:** Gut microbiota and metabolic alterations following increased protein and reduced carbohydrate diets.

Study Type	Bacterial Type	Metabolic Effects	References
**2-week high-protein/low-CHO diet in healthy older women (aged >65)**	Lactobacillus↑ Lactococcus↑ Streptococcus↑ Roseburia↓ Anaerostipes↓	Fat-free mass↑	[236]
**Hypocaloric high-protein/low-CHO diet with Bifidobacterium and lactobacillus synbiotic**	Bifidobacteria↑ Lactobacilli↑	Body weight↓ Fat mass↓ Waist Circumference↓ HbA1c↓	[237]
**High-protein/low-CHO vs. medium to high-CHO diet in obese humans**	Bifidobacteria↓ Roseburia↓ Eubacterium rectale↓	Butyrate↓	[239]
**8-week high-fat/low-CHO vs. low-fat/high-CHO diet in overweight and obese humans**	Bifidobacteria↓	SCFAs↓	[238]
**Crossover 4-week high-protein/low-CHO vs. high-protein/medium-CHO diet in obese humans**	Roseburia↓ Eubacterium rectale↓	BCFAs↑ Butyrate↓	[148]
**High-protein (55% vs. 30% vs. 14%) isocaloric diets in C57BL/6 Dextran Sodium Sulfate (DSS)-treated mice**	Proteobacteria↑ Actinobacteria↑ Bacteroidetes↑ Clostridium XIVa↓ Faecalibacterium↓ Roseburia↓	IL-6↑ IL-1β↑ Butyrate-producing genera↓	[244]

↓ indicates decrease; ↑ indicates increase; ↔ indicates no change. HbA1c: Glycated hemoglobin, SCFAs: Short-chain fatty acids, BCFAs: Branched-chain fatty acids, IL-6: Interleukin-6, IL-1β: Interleukin-1β.

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
