# Peer review of "Impact of Protein Intake in Older Adults with Sarcopenia and Obesity: A Gut Microbiota Perspective"

_nutrients, 2020, doi:10.3390/nu12082285_

Round 1

Reviewer 1 Report

Dear authors, 

Thank you for letting me participate in the review of your manuscript "Impact of Protein Intake in Older Adults withSarcopenia and Obesity: A Gut Microbiota Perspective" (nutrients-879100) which was submitted to the Journal Nutrients. 

It is a very interesting read and complete review on dietary protein intake on potential health impacts maybe related to changes of the composition of gut microbes. There were a few minor inconsistencies of the references (see attached PDF file) and in my opinion no other major revisions required. An interesting topic to discuss though would be microbial impacts on health which depend on the local environment (e.g. Yan He et. al. Regional variation limits applications of healthy gut microbiome reference ranges and disease models, Nature Medicine Oct 2018)

Reviewer 2 Report

  1. All the references should be in the prescribed journal format. Few references are incomplete (for example ref. 155). reference numbering in the text and reference section is not matching.
  2. Line number 38-40, check the reported data with recent relevant references. Please do check for the whole manuscript
  3. in the introduction part, only sarcopenia and obesity are described. All the content should be introduced in this section briefly to understand clearly the importance of this article.
  4. Line number 128-131, "A recent.....administration", is this a recent? references cited are not really recent. 
  5. Line 136, Faith et al., 2013 should be cited as journal format. do check for others like this (for example, line number 140,148,155)
  6. The conclusion should be concise and specific. Generally, references are not to be included in the conclusion. Its author's own perception and final massage for the whole manuscript.

Round 2

Reviewer 2 Report

Authors have revised the manuscript and replied the comments satisfactorily